# Vitamin D and Metabolic Dysfunction-Associated Fatty Liver Disease (MAFLD): An Update

**DOI:** 10.3390/nu12113302

**Published:** 2020-10-28

**Authors:** Ilaria Barchetta, Flavia Agata Cimini, Maria Gisella Cavallo

**Affiliations:** Department of Experimental Medicine, Sapienza University, Viale Regina Elena 321, 00161 Rome, Italy; ilaria.barchetta@uniroma1.it (I.B.); flaviaagata.cimini@uniroma1.it (F.A.C.)

**Keywords:** NAFLD, NASH, MAFLD, vitamin D, VDR, adipose tissue, gut, microbiota, inflammation, supplementation

## Abstract

Non-alcoholic fatty liver disease (NAFLD) is the first cause of chronic liver disease worldwide; it ranges from simple steatosis to steatohepatitis (NASH) and, potentially, cirrhosis and hepatocarcinoma. NAFLD is also an independent risk factor for type 2 diabetes, cardiovascular diseases, and mortality. As it is largely associated with insulin resistance and related disorders, NAFLD has been recently re-named as Metabolic dysfunction-Associated Fatty Liver Disease (MAFLD). At present, there are no approved pharmacological treatments for this condition. Vitamin D is a molecule with extensive anti-fibrotic, anti-inflammatory, and insulin-sensitizing properties, which have been proven also in hepatic cells and is involved in immune-metabolic pathways within the gut–adipose tissue–liver axis. Epidemiological data support a relationship hypovitaminosis D and the presence of NAFLD and steatohepatitis (NASH); however, results from vitamin D supplementation trials on liver outcomes are controversial. This narrative review provides an overview of the latest evidence on pathophysiological pathways connecting vitamin D to NAFLD, with emphasis on the effects of vitamin D treatment in MAFLD by a nonsystematic literature review of PubMed published clinical trials. This article conforms to the Scale for Assessment of Narrative Review Articles (SANRA) guidelines. Evidence so far available supports the hypothesis of potential benefits of vitamin D supplementation in selected populations of NAFLD patients, as those with shorter disease duration and mild to moderate liver damage.

## 1. Introduction

Non-alcoholic fatty liver disease (NAFLD) is the most common chronic liver disease worldwide and its evolution and consequences massively affect health and economic systems of Western countries [1,2]. NAFLD pathogenesis is primarily linked to metabolic impairment and alteration of the glucose–insulin homeostasis. For the tight connection between metabolic diseases and NAFLD, this condition has been recently re-named as Metabolic (dysfunction)-Associated Fatty Liver Disease (MAFLD) [3]; the new acronym MAFLD will be used to replace the term NAFLD throughout this review.

As a vicious circle, once MAFLD is established it increases the hepatic insulin resistance, which, in turn, may trigger, in 30–40% of cases, MAFLD evolution towards steatohepatitis (NASH), and eventually, cirrhosis, liver failure, and hepatocarcinoma [4,5]. MAFLD also promotes systemic low-grade inflammation and impairs insulin sensitivity in extra-hepatic tissues [6]. Finally, MAFLD increases the risk of type 2 diabetes (T2D) and diabetes’ complications and is an established risk factor for cardiovascular morbidity and mortality [7]. Data from four European countries, i.e., France, Germany, Italy, and UK, show that, in 2016, there were about 52 million people with MAFLD, with annual direct medical costs of about €35 billion, which were highest in patients in working age [8]. Remarkably, although MAFLD is already considered as the most rapidly growing contributor to liver mortality and morbidity [9], many reports and real world evidence show that MAFLD is still an under-recognized and under-diagnosed condition in both primary and secondary care [9,10,11]. These data point towards urgent need of better risk stratification, earlier diagnosis, and management of MAFLD in order to decrease the short- and long-term public health burden of this disease. At present, there are no approved pharmacological treatments for MAFLD or steatohepatitis [9].

Vitamin D is a pleiotropic hormone with functions that extend far beyond the regulation of calcium homeostasis and bone mineralization; in the last decades, experimental evidence has definitively proven the involvement of vitamin D in mediating a number of immune-inflammatory [11] and metabolic [12] processes. Since then, the axis involving the active form of vitamin D- 1,25-dihydroxi-vitamin D- and the vitamin D receptor (VDR) has been investigated in relation to disturbances of metabolic pathways in several organs and tissues, primarily in those implicated in metabolic regulation, as the skeletal muscle [13], adipose tissue [14], pancreas [15], and liver [16,17,18]. The presence of hypovitaminosis D has been associated to the occurrence or development of insulin resistance-related diseases, such as T2D [19], metabolic syndrome [20], and MAFLD [16,18,21,22,23]. As for vitamin D and liver diseases, in the last years several clinical trials tried to answer the question whether vitamin D supplementation could improve MAFLD, with controversial results.

This review aims to provide an overview of the most recent evidence on pathophysiological pathways connecting the vitamin D/VDR axis to MAFLD development and will focus on new data from clinical trials exploring the safety and efficacy of vitamin D supplementation on liver outcomes in individuals with MAFLD.

This narrative review provides an overview of the latest evidence on pathophysiological pathways connecting vitamin D to NAFLD and describes results from a nonsystematic literature review of published clinical trials on vitamin D treatment in MAFLD. The reporting of this study conforms to the Scale for Assessment of Narrative Review Articles (SANRA) guidelines, a brief critical appraisal for the assessment of nonsystematic articles [24].

A literature search was performed up to October 2020 with database of Pubmed. “Non-alcoholic fatty liver disease” (NAFLD), “non-alcoholic steatohepatitis” (NASH), “fatty liver”, “hepatic steatosis”, “metabolic-associated fatty liver disease” (MAFLD) were paired with “vitamin D”, “cholecalciferol”, “calcitriol”, “ and vitamin D receptor” (VDR) as search terms. Moreover, manual search was also conducted by scrutinizing the reference lists of original articles, meta-analyses, and recent reviews. Inclusion criteria to identify relevant studies were: studies conducted worldwide on adult humans aged ≥18 years with a diagnosis of MAFLD treated with oral or bolus cholecalciferol or calcitriol supplementation in comparison to MAFLD individuals supplemented with placebo. Pilot studies without comparators were also included. Outcome measures were changes of (i) hepatic fat content, as estimated by abdomen ultrasound (US), magnetic resonance imaging (MRI), magnetic resonance spectroscopy (MRS), and liver biopsy; and (ii) hepatic enzymes (aspartate aminotransferase (AST), alanine aminotransferase (ALT), and gamma glutamyl transpeptidase (GGT)), fibrosis markers and scores (cytokeratin-18 (CK-18), procollagen III amino terminal propeptide (PIIINP), Fibrosis-4 Index for Liver Fibrosis (FIB-4), and Enhanced Liver Fibrosis (ELF) score). Study design was randomized clinical trial (RCT) with parallel or cross-over design. Articles published in languages other than English were excluded. For each study, we report surname of first author, published year, region/nation, sample-size and control to intervention ratio, endpoints, types of intervention, doses, and duration of intervention. Data are presented in the manuscript and summarized in a table.

## 2. Vitamin D/VDR Axis in the Pathophysiology of MAFLD

The potential involvement of the vitamin D/VDR axis in the pathogenesis and progression of MAFLD has been suggested by experimental studies linking vitamin D-mediated pathways to key processes leading to liver steatosis, inflammation, and fibrosis. Indeed, vitamin D may influence MAFLD development in both direct and indirect manner [25]. Figure 1 summarizes potential pathways linking vitamin D/VDR axis to the development of MAFLD.

Vitamin D and liver homeostasis: Vitamin D displays systemic and tissue-specific anti-inflammatory properties [26] that have been experimentally demonstrated also at the hepatic level. In MAFLD rat models, treatment with active vitamin D reduced liver inflammation and oxidative stress by inhibiting the p53-p21 signaling pathway and associated cell senescence [26]; vitamin D also protected against high fat diet-induced fatty liver by promoting the nuclear translocation of the anti-oxidant molecule nuclear factor erythroid 2-related factor 2 (NFE2L2) [27], decreasing toll-like receptors [28] or repressing sirtuin [29]. Furthermore, in a recent investigation, vitamin D improved hepatic insulin resistance and ameliorated liver steatosis in rodent models via the VDR-mediated activation of the hepatocyte nuclear factor 4α (HNF4α) [30].

On the other side, vitamin D deficiency exacerbates liver inflammation [28]. In humans, hepatic VDR expression inversely correlated with steatosis severity and lobular inflammation at the liver histology [31]. In addition, the activation of VDRs in hepatic macrophages by vitamin D ligands ameliorated liver inflammation, steatosis, and insulin resistance in experimental studies [32].

Vitamin D exerts anti-fibrotic activity in the liver by inhibiting the proliferation of hepatic stellate cells and the expression of pro-fibrotic mediators such as the platelet-derived growth factor (PDGF) and the transforming growth factor β (TGF-β); similarly, vitamin D suppresses the expression of collagen, α-smooth muscle actin and tissue inhibitors of metalloproteinase-1 [16,33]. Mice knocked-out for the VDR gene spontaneously developed hepatic damage and fibrosis toward to frank cirrhosis [34]. However, active vitamin D administration fails to ameliorate experimentally-induced liver damage in animal models when cirrhosis is already established [35].

Finally, recent clinical evidence pointed towards a role of liver VDR expression in modulating intra-hepatic lipid accumulation, potentially by controlling the local levels of the angiopoietin-like protein 3 and lipoprotein-lipase [36].

In addition to exerting direct insulin-sensitizing, anti-inflammatory, and anti-fibrotic actions in the liver parenchyma, the vitamin D/VDR system participates also to the maintenance of systemic insulin sensitivity and to the homeostasis of organs involved in MAFLD pathogenesis, such as the gut and the adipose tissue [37,38,39,40,41].

Vitamin D and insulin sensitivity: The vitamin D/VDR axis regulates metabolic pathways associated with insulin sensitivity and glucose-insulin homeostasis [42,43,44], favors glucose uptake in muscle cells by upregulating the intracellular expression of the insulin receptor substrate IRS-1 [45] and promotes the expression of the insulin-dependent glucose transporter 4 (GLUT-4) on adipocytes [14]. Moreover, the insulin gene is transcriptionally regulated by VDR in pancreatic β cells, and vitamin D deficiency reduces insulin secretory response to carbohydrate loading in experimental models [46]. Indeed, vitamin D insufficiency has been independently associated with insulin resistance in overweight and obese individuals [20], fostering the risk of MAFLD development.

Vitamin D and adipose tissue (AT): Among several pathophysiological processes occurring in presence of chronically excessive caloric intake and weigh gain, the alterations of the AT structure and function represent major determinants of MAFLD development in obesity [37]. In these conditions, AT dysfunction is induced by loss of adipocyte plasticity, insufficient storing capacity, with subsequent matrix rearrangement, hypoxia, and inflammatory reaction [38,39]. In turn, the increased release of fatty acids, pro-inflammatory cytokines, and adipokines [47,48,49,50] from stressed adipocytes into the bloodstream leads to low-grade chronic inflammation and abnormal fat deposition in ectopic sites, mostly into the liver [51].

Signatures of AT dysfunction and metabolic impairment are associated to increased intra-hepatic fat accumulation across different body mass index (BMI) classes, in individuals with and without T2D [38,48,52].

In addition to representing the main storage site for vitamin D and expressing key enzymes involved in vitamin D metabolism, AT is also a primary target of vitamin D action, where this hormone modulates insulin-sensitivity, local inflammation, and adipokine secretion. Evidence from clinical [53] and experimental studies [54] showed that treatment with vitamin D improved AT oxidative stress [53] and local concentrations of pro-inflammatory cytokines, such as the tumor necrosis factor α TNF-α and the monocyte chemoattractant protein-1 (MCP-1) [54]. Indeed, vitamin D ameliorates AT inflammation and prevents liver steatosis by reducing both AT output of lipid droplets and hepatic de novo lipogenesis and fatty acid oxidation [55]. Moreover, treatment with calcitriol increases VDR expression in peripheral cells, ameliorates systemic and tissue-associated inflammatory profile, reducing AT inflammation and liver steatosis in animal models [56]. In the AT, the vitamin D/VDR axis influences both adipogenesis and lipid storage into the adipocytes [57]. Of note, AT VDR expression levels are increased in human [58] and experimentally-induced [57] obesity independently from overall vitamin D status. Greater VDR expression may potentially represent a compensatory response to impaired local activation and/or action in condition of altered AT homeostasis. Differential VDR expression in visceral AT may also influence lipid storage and adipocytes enlargement though the transcriptional regulation of angiopoietin-like protein 4 and lipoprotein lipase, which in turn result in liver impairment in obesity [59].

Vitamin D and gut homeostasis: Vitamin D is centrally involved in the regulation of the gut-adipose tissue–liver axis, which represents a major pathway leading to metabolic complications in obesity. In particular, evidence from the last decade underpins a major role of the gut–liver crosstalk in the pathogenesis of MAFLD [37]. Indeed, the gut is nowadays considered as a promising target for experimental therapies of liver steatosis [60,61]. Of note, in humans the gastrointestinal tract is a major site of expression of VDR [62,63] where this receptor mediates the vitamin D action in preserving the gut homeostasis [61] via different regulatory activities such as immuno-modulation [64,65], preservation of the barrier function [66] and regulation of the gut microbiota [67,68].

As an immune adjuvant, vitamin D maintains immune tolerance in the gut microenvironment by suppressing adaptive immunity and up-regulating innate immunity [69]. Indeed, vitamin D inhibits T helper (Th-) 1 and enhances Th2 cell responses; it also decreases Th17 cell differentiation, and increases regulatory T (TReg) cells [70,71]. Th1, Th2, and Th17 cells are known to cause mucosal inflammation and tissue injury, whereas Tregs, which are important intermediaries of immune tolerance, play anti-inflammatory functions and mitigate mucosal inflammation and stimulate tissue repair [72,73,74,75].

Several studies showed that VDR regulates the expression of the tight junctions zona occludens proteins 1 and 2 (ZO-1 and ZO-2) through the up-regulation of claudin 2 and 12 and the down-regulation of cadherin-17 [76,77] so preserving the adhesive phenotype of intestinal epithelial cells. Moreover, vitamin D repairs tight junctions upon bacterial lipopolysaccharide-mediated injury [78] and maintains mucosal barrier integrity by preventing intestinal cell apoptosis in inflammation [79,80]. Mice knocked-out for the intestine VDR gene and those with vitamin D deficiency, displayed alterations of the tight junctions and increased gut permeability [47]. Vitamin D replacement prevents the development of experimentally-induced steatosis, inflammation and fibrosis [81] and restores gut dysbiosis [82].

Indeed, several studies demonstrated that vitamin D/VDR axis is involved in the modulation of gut microbiota [82,83,84,85,86,87,88,89,90,91], which in turn impacts on the development of MAFLD in obesity [41]. The genetic ablation of VDR in mice induces gut dysbiosis, reducing *Lactobacillus* and increasing *Clostridium* and *Bacteroides* concentrations [86].

These overall findings warrant further studies evaluating the modulation of vitamin D/VDR signaling as a future therapeutic approach to MAFLD.

## 3. Vitamin D Supplementation and MAFLD: Evidence from Clinical Trials

The observations regarding the link between hypovitaminosis D and the presence of MAFLD suggest that vitamin D supplementation might represent a potential therapeutic option for MAFLD in both children [92,93] and adult populations [18,94,95].

However, data from meta-analyses have not confirmed univocally the presence of a relationship between hypovitaminosis D and MAFLD, especially in trials where histological-, rather than clinical-, or biochemical-outcomes of liver damage were considered [96,97].

Similarly, controversial findings result from interventional clinical trials investigating the efficacy of vitamin D supplementation on parameters of MAFLD and steatohepatitis [98,99] Table 1.

The first pilot study on NASH was conducted by Kitson MT et al. [100] in patients with NASH undergoing liver biopsy before and after six-month 25,000 IU cholecalciferol weekly supplementation and showed no effect on liver outcomes as local inflammation, fibrosis, and intrahepatocyte fat accumulation [100]. Improvement of non-specific clinical and biochemical markers of liver damage after vitamin D supplementation was found in some clinical trials [101,102,103], whereas no vitamin D effect on the same parameters was shown in other reports [104,105,106].

In 2016 our group published the results from the first randomized, double-blind, placebo-controlled clinical trial conducted in individuals with T2D and MAFLD, where intrahepatic fat content was measured by magnetic resonance imaging (MRI) [106]. In this study, participants underwent 24-week high-dose oral cholecalciferol supplementation (2000 IU a day) and no effect was shown on either hepatic fat percentage or markers of hepatic injury and/or fibrosis, i.e., serum transaminases, CK-18 and PIIINP levels. Moreover, no beneficial effect was reported for any metabolic parameter, such as body adiposity, glycemic control, estimated insulin resistance, blood pressure or endothelial dysfunction [106].

Conversely, results from a clinical trial conducted by Geier et al. [107] showed that 48-week vitamin D3 treatment (2100 IU vitamin D3 daily) leads to significantly decreased serum ALT and CK-18 levels in twenty individuals with biopsy-proven NASH. Pre- to post-intervention histological changes were investigated in a sub-cohort of seven individuals, finding no significant modification [106].

Dabbaghmanesh et al. [108] published results from a randomized, double blind, placebo controlled trial investigating the effect of three-month high dose oral vitamin D3 (50,000 IU/week) or calcitriol (0.25 mg per day) supplementation in over 100 non-diabetic vitamin D deficient individuals with ultrasound-diagnosed NAFLD and normal transaminases. Neither vitamin D3 nor calcitriol supplementation significantly modified liver enzymes in comparison to placebo; no data on liver fat content and/or indirect indexes of hepatic fibrosis were available in this study [108].

A similar investigation was conducted by Naderpoor et al. [109] aiming to explore the effects of vitamin D supplementation (100,000 loading dose of cholecalciferol followed by 4000 IU daily for 16 weeks) on liver enzymes in 54 overweight/obese individuals with vitamin D deficiency at the time of the study enrolment and no history of liver disease, without finding any change [109].

A recent randomized, double-blind, placebo-controlled trial explored the effect of vitamin D supplementation (3200 IU daily for three months) on cardiovascular risk factors, hormones, and liver markers in women with polycystic ovary syndrome, finding modest improvement of ALT and enhanced liver fibrosis (ELF) score, along with a trend towards reduced insulin-resistance-estimated by the Homeostatic Model Assessment for Insulin Resistance (HOMA-IR) index—in the actively treated group versus placebo [110].

Overall, clinical trials conducted so far have adopted non-homogenous inclusion criteria, recruited population with broad range of MAFLD severity and selected non-comparable outcome measures; this inhomogeneity has inevitably led to conflicting results.

## 4. Potential Pitfalls and Future Directions

Vitamin D and VDR regulate numerous mechanisms associated with inflammatory responses, insulin function and overall immune-metabolism. Accumulating data from mechanistic studies confirm a major involvement of vitamin D in both liver homeostasis itself and in the gut–adipose tissue–liver axis, and many observational studies have reported the existence of a close relationship between hypovitaminosis D and MAFLD. However, some other investigations have not found any correlation between vitamin D status and liver damage at the histological examination.

In the last years, several clinical trials have tested potential benefits of vitamin D on MAFLD in in cohorts of individuals with different extension of liver damage, comorbidities and comedications, producing inconsistent findings. The latest meta-analysis which has explored the effects of vitamin D supplementation on cardio-metabolic and hepatic outcomes in patients with MAFLD, including data from almost 550 participants [99], concluded that vitamin D supplementation may have beneficial effects on glucose–insulin metabolism, ALT, and triglycerides in the youngest sub-group of patients (<45 years old) [99]. Moreover, based on the clinical trials published so far, longer exposure to vitamin D dose less than 3500 IU per day, seems to exert the best effect on transaminases reduction in MAFLD patients [99].

In the light of the available evidence, a major role behind successful vitamin D supplementation on liver parameters is played by the improvement of blood glucose and insulin levels [100,101,102,103,104,105,106,107,108,109,110,111,112]. Indeed, in individuals without established diabetes and antidiabetic therapies, vitamin D may positively impact on glucose tolerance and insulin resistance and reduce directly and indirectly liver impairment in MAFLD. Conversely, once diabetes is established, vitamin D may not be sufficient to remodulate glucose homeostasis and/or its potential effects may be hidden by concomitant therapies. Indeed, in dysmetabolic individuals undertaking multiple treatments, as those with frank metabolic syndrome and diabetes, a certain effect of comedications on liver status, as intended as either damage-in addition to MAFLD itself, or potential benefit as for some new antidiabetic agents [113], cannot be definitively ruled out.

Differently from what reported in terms of potential benefit of vitamin D supplementation on glucose–insulin profile, evidence of efficacy on liver fibrosis and inflammation is still lacking.

These overall findings may support the hypothesis that vitamin D supplementation may exert beneficial effects mostly in younger individuals, with shorter disease duration and mild to moderate liver damage [114] and/or in addition to anti-fibrotic agents [115].

## 5. Conclusions

Convincing experimental data show that the vitamin D/VDR axis is directly involved in the modulation of metabolic and inflammatory pathways associated with the development of MAFLD in overweight and obesity. Indeed, vitamin D and VDR take part not only in intra-hepatic regulation of insulin sensitivity, fat accumulation, and immune-inflammatory responses, but also in the homeostasis of organs that are primarily involved in the pathogenesis of NAFLD and NASH, such as gut and adipose tissue.

Clinical trials do not report unequivocal beneficial effects of vitamin D supplementation on markers of liver impairment in individuals with MAFLD. Moreover, the investigations conducted so far involved small populations and were markedly heterogeneous in terms of inclusion criteria, study design and outcome measures.

Nonetheless, the evidence available shows positive effects of long-term low-dose vitamin D treatment in the youngest populations of MAFLD subjects, without hepatic fibrotic damage and clinically overt complications and comorbidities, such as T2D. Vitamin D is a molecule with beneficial effects on a large number of organs and systems, primarily the skeleton and immune system [11,12,13], and its supplementation is considered a highly cost-effective strategy for disease prevention, i.e., fractures’ risk reduction [116].

Further studies on larger populations of individuals, selected in relation to criteria emerging from available clinical trials, may be needed before drawing general conclusions on the benefit of vitamin D supplementation in patients with fatty liver disease.

## Figures and Tables

**Figure 1 nutrients-12-03302-f001:**
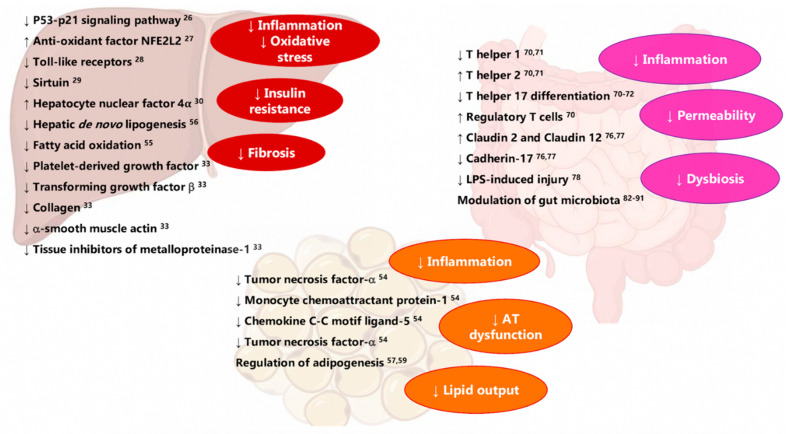
Potential pathways linking vitamin D/vitamin D receptor (VDR) axis to Metabolic dysfunction-Associated Fatty Liver Disease (MAFLD). AT—adipose tissue; ↑—increase; ↓—decrease.

**Table 1 nutrients-12-03302-t001:** Characteristics of clinical trials evaluating vitamin D supplementation on MAFLD. Abbreviations: US—Ultrasound, MRI—Magnetic Resonance Imaging, ELF—Enhanced Liver Fibrosis score, IU—International Units.

Author	Year	Country	No. (Control/Intervention) Endpoint	Duration	Dose of Vitamin D	Type of Intervention		Results
Foroughi M. [104]	2014	Iran	60 (30/30)	Hepatic steatosis (US), enzymes	10 weeks	50,000 IU per week	Vitamin D3	No effect
Sharifi N. [105]	2014	India	53 (26/27)	Hepatic enzymes	4 months	50,000 IU per 14 days	Vitamin D3	No effect
Kitson M.T. [100]	2016	Australia	12	Liver histology	24 weeks	25,000 IU per week	Vitamin D3	No effect
Lorvand Amiri H. [102]	2016	Iran	120 (36/74)	Hepatic enzymes	12 weeks	1000 IU day	Calcitriol	Significant effect
Lorvand Amiri H. [103]	2016	Iran	73 (36/37)	Hepatic steatosis (US)	12 weeks	1000 IU day	Calcitriol	Significant effect
Barchetta I. [106]	2016	Italy	55 (29/26)	Intrahepatic fat content (MRI)	24 weeks	2000 IU day	Vitamin D3	No effect
Sakpal M. [101]	2017	Iran	81 (30/51)	Hepatic enzymes	6 months	600,000 IU i.m./6 months	Vitamin D3	Significant effect
Geier A. [107]	2018	Switzerland	18 (10/8)	Liver histology	48 weeks	2100 IU day	Vitamin D3	No effect
Dabbaghmanesh M.H. [108]	2018	Iran	63 (32/31)	Hepatic enzymes	12 weeks	50,000 IU per week	Vitamin D3	No effect
Naderpoor N. [109]	2018	Australia	54 (28/26)	Hepatic enzymes	16 weeks	4000 IU day	Vitamin D3	No effect
Javed Z. [110]	2019	UK	37 (18/19)	Hepatic enzymes, ELF score	3 months	3200 IU day	Vitamin D3	Significant effect

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
