# Peer review of "Vitamin D and Metabolic Dysfunction-Associated Fatty Liver Disease (MAFLD): An Update"

_nutrients, 2020, doi:10.3390/nu12113302_

Round 1
Reviewer 1 Report
The authors present a succinct and relevant review on the role of vitamin D in MAFLD. The authors should consider seeking some grammar and syntax aid, as several subject-verb agreement issues, misused words, and run-on sentences were present.
Line 35: Good information, however most individuals who have NAFLD (MAFLD) never see it progress to NASH. Should address.
Line 40: most reports state our current estimations of NAFLD (MAFLD) are low because of how we screen and test for it. Recommend adding a line about this to drive home the potential dire consequences we could be facing.
Line 47: need to spell out vitamin D receptor before abbreviating. Please check for other instances of using abbreviations without first defining.
Line 155: nice job covering the various pathways associated with vitamin D and MAFLD. Recommend the authors include a figure to visually represent these pathways.
Line 201: Recommend authors include a table summarizing studies and the studies' major findings.
Author Response
Response to Reviewer 1
The authors present a succinct and relevant review on the role of vitamin D in MAFLD. The authors should consider seeking some grammar and syntax aid, as several subject-verb agreement issues, misused words, and run-on sentences were present.
The manuscript has been carefully checked and we corrected all the mistakes that we could find.
Line 35: Good information, however most individuals who have NAFLD (MAFLD) never see it progress to NASH. Should address.
We thank the reviewer for this comment and have now specified that the progression towards NASH only occurs in a subgroup of NAFLD individuals: “may trigger, in 30-40% of cases, MAFLD evolution towards steatohepatitis (NASH), and eventually, cirrhosis, liver failure and hepatocarcinoma” (page 1, lines 39-40; new reference 4).
Line 40: most reports state our current estimations of NAFLD (MAFLD) are low because of how we screen and test for it. Recommend adding a line about this to drive home the potential dire consequences we could be facing.
We thank the reviewer for this comment which allowed us to address this important point in the revised version of our manuscript: “Remarkably, although MAFLD is already considered as the most rapidly growing contributor to liver mortality and morbidity [9], many reports and real world evidence show that MAFLD is still an under-recognized and under-diagnosed condition in both primary and secondary care [9-11]. These data point towards urgent need of better risk stratification, earlier diagnosis and management of MAFLD in order to decrease the short- and long- term public health burden of this disease”; page 2, lines 46-50, new references 9 and 10).
Line 47: need to spell out vitamin D receptor before abbreviating. Please check for other instances of using abbreviations without first defining.
“Vitamin D receptor” and other expressions have now been spelt out before abbreviating in the manuscript.
Line 155: nice job covering the various pathways associated with vitamin D and MAFLD. Recommend the authors include a figure to visually represent these pathways.
We thank the reviewer and have now added a figure (Figure 1, page 3) which illustrates pathways associating vitamin D and MAFLD.
Line 201: Recommend authors include a table summarizing studies and the studies' major findings.
A table summarizing vitamin D supplementation studies and their major findings has now been included in the new version of this manuscript (Table 1, page 6).
Reviewer 2 Report
Abstract
Please describe here what sort of review you have done. Is this a systematic review? Also, there is a need to the brief description of method for search and also the results of search. Moreover, please add some details of the intitial results and the relevent conclusion.
Body of the manuscript
Based on the introduction, I found that your article is mainly a discussion paper rather than a review. Therefore, throughout the text, this should be recognised. If you believe that a review have been performed, you should reorganise your article using https://www.equator-network.org/reporting-guidelines/prisma/.
You are recommended to add a conclusion section and present implications of your discussion for future research, practice, policy making etc.
Author Response
Response to Reviewer 2
Abstract:
Please describe here what sort of review you have done. Is this a systematic review? Also, there is a need to the brief description of method for search and also the results of search. Moreover, please add some details of the initial results and the relevant conclusion.
We have now specified in the abstract and manuscript that, for its characteristics, this article represents a narrative review (Higgins JPT, Green S. (eds.). Cochrane Handbook for Systematic Reviews and Interventions. Version 5.1.0. The Cochrane Collaboration 2011, section 1.2.2., www.handbook.cochrane.org, retrieved on Oct 31, 2018). A brief description of search methods, along with details of initial results and most relevant conclusions have been added in the abstract, as suggested by the reviewer: “This narrative review provides an overview of the latest evidence on pathophysiological pathways connecting vitamin D to NAFLD, with emphasis on the effects of vitamin D treatment in MAFLD by a nonsystematic literature review of PubMed published clinical trials. This article conforms to the Scale for Assessment of Narrative Review Articles (SANRA) guidelines. Evidence so far available supports the hypothesis of potential benefits of vitamin D supplementation in selected populations of NAFLD patients, as those with shorter disease duration and mild to moderate liver damage”, page 1 lines 20-26.
Body of the manuscript:
Based on the introduction, I found that your article is mainly a discussion paper rather than a review. Therefore, throughout the text, this should be recognised. If you believe that a review have been performed, you should reorganise your article using https://www.equator-network.org/reporting-guidelines/prisma/.
We thank the reviewer for this comment and have now specified that this article is a narrative review. For this reason, the manuscript was not organized according to the PRISMA guidelines which are applicable only to systematic reviews. Although this kind of validated critical appraisal and quality assessment tool is not available for narrative review, we Nonetheless, in line with this reviewer’s suggestion, we have now provided information on the methods for search adopted in a new paragraph of the manuscript: “This narrative review provides an overview of the latest evidence on pathophysiological pathways connecting vitamin D to NAFLD and describes results from a nonsystematic literature review of published clinical trials on vitamin D treatment in MAFLD. Our PubMed search terms included NAFLD, NASH, MAFLD pathophysiology, treatment, combined with words vitamin D', 'cholecalciferol’, ‘calcitriol’, vitamin D receptor (VDR)’. The reporting of this study conforms to the Scale for Assessment of Narrative Review Articles (SANRA) guidelines, a brief critical appraisal for the assessment of nonsystematic articles [24]”; page 2, lines 67-73; new reference 24.
You are recommended to add a conclusion section and present implications of your discussion for future research, practice, policy making etc.
A conclusion section has now been added to the manuscript, as suggested by this reviewer (Paragraph 5, page 8, lines 269-285).
Round 2
Reviewer 2 Report
Dear Authors,
Therefore, you have made a decision on the method's section as a narrative review. Please be informed that even if you have done a narrative review there must be a brief description for the review process in terms of databases, year, inclusion criteria and how the findings have been summarised and presented. Therefore, make this important change to be able to call your writing as a narrative review. If not, I do recommend you to change the method to a discussion as it does not need any special description of the methods section and completely suits the present style of presentation.
Author Response
Dear Authors, Therefore, you have made a decision on the method's section as a narrative review. Please be informed that even if you have done a narrative review there must be a brief description for the review process in terms of databases, year, inclusion criteria and how the findings have been summarised and presented. Therefore, make this important change to be able to call your writing as a narrative review. If not, I do recommend you to change the method to a discussion as it does not need any special description of the methods section and completely suits the present style of presentation.
We thank the reviewer for this comment and have now added a brief description of the review process in the revised version of this manuscript: “A literature search was performed up to October 2020 with database of Pubmed. ‘Non-alcoholic fatty liver disease’ ( NAFLD), ‘non-alcoholic steatohepatitis’ (NASH), ‘fatty liver’, ‘hepatic steatosis’, ‘metabolic-associated fatty liver disease’ (MAFLD) were paired with ‘vitamin D', 'cholecalciferol’, ‘calcitriol’, vitamin D receptor (VDR)’ as search terms. Moreover, manual search was also conducted by scrutinizing the reference lists of original articles, meta-analyses and recent reviews. Inclusion criteria to identify relevant studies were: studies conducted worldwide on adult humans aged ≥18 years with a diagnosis of MAFLD treated with oral or bolus cholecalciferol or calcitriol supplementation in comparison to MAFLD individuals supplemented with placebo. Pilot studies without comparators were also included. Outcome measures were changes of: i) hepatic fat content, as estimated by abdomen ultrasound (US), magnetic resonance imaging (MRI), magnetic resonance spectroscopy (MRS), liver biopsy, ii) hepatic enzymes (AST, ALT, GGT), fibrosis markers and scores (CK-18, PIIINP, FIB-4, ELF). Study design was randomized clinical trial (RCT) with parallel or cross-over design. Articles published in languages other than English were excluded. For each study, we report surname of first author, published year, region/nation, sample-size and control to intervention ratio, endpoints, types of intervention, doses, duration of intervention. Data are presented in the manuscript and summarized in a table”; page 2, lines 72-87.